# TANDEM BLOCKS IN DEEP CONVOLUTIONAL NEURAL NETWORKS

## ABSTRACT

Due to the success of residual networks (resnets) and related architectures, shortcut connections have quickly become standard tools for building convolutional neural networks. The explanations in the literature for the apparent effectiveness of shortcuts are varied and often contradictory. We hypothesize that shortcuts work primarily because they act as linear counterparts to nonlinear layers. We test this hypothesis by using several variations on the standard residual block, with different types of linear connections, to build small (100k–1.2M parameter) image classification networks. Our experiments show that other kinds of linear connections can be even more effective than the identity shortcuts. Our results also suggest that the best type of linear connection for a given application may depend on both network width and depth.

## 1 INTRODUCTION

Deep convolutional neural networks have become the dominant force for many image classification tasks; see Krizhevsky et al. (2012); Simonyan & Zisserman (2014); Szegedy et al. (2014). Their ability to assimilate low-, medium-, and high-level features in an end-to-end multi-layer fashion has led to myriad groundbreaking advances in the field. In recent years, residual networks (resnets) have emerged as one of the best performing neural network archetypes in the literature; see He et al. (2015). Through the use of identity shortcut connections, resnets have overcome the challenging technical obstacles of vanishing gradients and the apparent degradation that otherwise comes with training very deep networks. Resnets have achieved state-of-the-art performance on several image classification datasets using very deep neural networks, sometimes with over 1000 layers.

Although shortcut connections appeared in the early neural network literature, e.g., Bishop (1995); Ripley & Hjort (1995); Schraudolph (2012), their importance became more clear in 2015 with the emergence of the HighwayNets of Srivastava et al. (2015) and resnets. The former involved gated shortcut connections that regulate the flow of information across the network, while the latter used identity shortcut connections, which are parameterless. Resnets are also presumed to be easier to train and seem to perform better in practice. In their first resnet paper, He et al. argued that identity maps let gradients flow back, enabling the training of very deep networks, and that it's easier for a layer to learn when initialized near an identity map than near a zero map (with small random weights); see also He et al. (2016).

However, in a flurry of recent activity, most notably from Zagoruyko & Komodakis (2016); Greff et al. (2016); Veit et al. (2016); Li et al. (2016) and Wu et al. (2016), arguments have emerged that the effectiveness of resnets is not due to their depth, where practitioners were training networks of hundreds or thousands of layers, but rather that deep resnets are effectively creating ensembles of shallower networks, and the layers are more likely to refine and reinforce existing features than engineer new ones. These arguments assert that the achievement of resnets is less about extreme depth and more about their ability to ease backpropagation with moderate depth. Indeed, in many cases wider residual networks that were only 10–50 layers deep were shown to perform better and train in less time than very deep ones (over 100 layers). See Zagoruyko & Komodakis (2016).

More recently still, others have presented many clever and creative ways to train very deep networks using variations on the shortcut theme; see for example Huang et al. (2016); Larsson et al. (2016); Zhang et al. (2016a); Han et al. (2016); Abdi & Nahavandi (2016); Chen et al. (2017); Zhang et al. (2016b); He et al. (2015); Lee et al. (2017); Xie et al. (2016); Savarese (2016); Szegedy et al. (2014;

2015), and Szegedy et al. (2016). In summary, shortcut connections clearly help in practice, but there are many different, and sometimes conflicting hypotheses as to why.

In this paper we investigate a new hypothesis about shortcut connections, namely, that their power lies not in the identity mapping itself, but rather just in combining linear and nonlinear functions at each layer. The tests where identity shortcuts were observed to perform better than general linear connections were all done in very deep (100 or more layers) networks. The recent evidence that wider, shallower, resnet networks can outperform deeper ones suggests that it is worth investigating whether identity connections are better than general linear connections in such networks.

We first describe some of the intuition about why this might be the case. We then investigate this idea with careful experiments using relatively small networks constructed of five different types of blocks. These blocks are all variations on the idea of residual blocks (resblocks), but where the identity shortcut is replaced with a more general linear function. We call these blocks, consisting of both a linear and a nonlinear part, *tandem blocks* and the resulting networks *tandem networks*. Residual networks and several similar architectures are special cases of tandem networks.

The networks we use in our experiments are relatively small (100k–1.2M parameter) image classification networks constructed from these various tandem blocks. The small networks are appropriate because the goal of the experiments is not to challenge state-of-the-art results produced by much larger models, but rather to compare the five architectures in a variety of settings in order to gain insight into their relative strengths and weaknesses. Whereas many other authors pursue extreme network depth as a goal in itself, here we limit our focus to comparing performance (in this case, classification accuracy) of different architectures.

Our experiments suggest that general linear layers, which have learnable parameters, perform at least as well as the identity shortcut of resnets. This is true even when some width is sacrificed to keep the total number of parameters the same. Our results further suggest that the best specific type of linear connection to use in the blocks of a tandem network depends on several factors, including both network width and depth.

## 2 TANDEM BLOCKS

The basic building block we use is the *tandem block*, which consists of a linear and a nonlinear part in parallel (see Figure 1). In the case of resnet blocks, the linear part is simply an identity shortcut. But a tandem block generalizes this to allow any linear map. The outputs of the two parts are summed and then passed to subsequent blocks.

Note that, unlike in the case of the original resnet paper, we do not pass the resulting sum to another nonlinear activation function. This is because recent work of Dong et al. (2017) has shown that removing the activation function after the sum in resnets improves performance.

Many authors assert that identity shortcuts are superior to other linear maps in such a configuration. See He et al. (2016); Szegedy et al. (2016); He et al. (2015); Savarese (2016), and Li et al. (2016). The reasons given for this assertion vary and are usually heuristic in nature. We chose the architectures we did in order to test whether the following properties of identity shortcuts were important: having no parameters, maintaining feature size (as opposed to creating higher-level features with $3 \times 3$ filters), and bypassing multiple nonlinear convolutional layers.

### 2.1 BUILDING BLOCKS

We consider five different tandem blocks, each of which can be viewed as a variant on the standard resblock. The nonlinear part of each block (depicted on the right side of each block in Figure 1) consists of either one or two layers of activated (nonlinear) convolutions. The linear part of each block (depicted on the left in Figure 1) consists of either (i) identity maps, corresponding to standard resblocks or (ii) convolutions of size $1 \times 1$ or $3 \times 3$, corresponding to more general tandem blocks. Note that in some (usually rare) cases identity maps are impossible to use because of mismatches in layer width. In those cases, as is typical with resnets, we use $1 \times 1$ convolutions to project the identity to have the necessary width. In all cases the outputs of the linear and nonlinear parts are added together at the end.

Specifically, the block variants we defined are as follows:

- $B_{\mathrm{id}}(2, w)$ is the standard residual block, with two activated convolutional layers and an identity connection from start to finish. As is standard for resnets, in the relatively rare case that layers of different widths must be connected (and hence the identity cannot be used), the identity connection is replaced with a more flexible $1 \times 1$ convolution.

- $B_{\mathrm{id}}(1, w)$ is the same as $B_{\mathrm{id}}(2, w)$, but with only one activated convolution instead of two. This is another form of resblock.

- $B_{1 \times 1}(2, w)$ is a tandem block like $B_{\mathrm{id}}(2, w)$ except that it always uses $1 \times 1$ convolutions instead of identity connections, even when connecting layers of the same width.

- $B_{1 \times 1}(1, w)$ is a tandem block like $B_{1 \times 1}(2, w)$, but with only one activated convolution instead of two.

- $B_{3 \times 3}(1, w)$ has $3 \times 3$ convolutions on both sides, but on the nonlinear side it is followed by a nonlinear activation function while on the linear side it is not.

These are all shown in Figure 1. The first four blocks have all been used in previous publications, such as He et al. (2016). We believe that the use of $3 \times 3$ filters for linear connections is unique. However, our primary goal is not to introduce a novel architecture. Rather, it is to investigate *why* tandem blocks, including residual blocks, work. We chose this selection of blocks in order to determine which properties of shortcut connections (identity maps, fixed weights, linearity, etc) were necessary and which were unnecessary or even suboptimal. In particular, these blocks were ideal for determining whether learnable convolutions are just a necessary evil for accommodating occasional changes in layer width or whether they are a viable, or even superior, alternative to identity shortcuts.

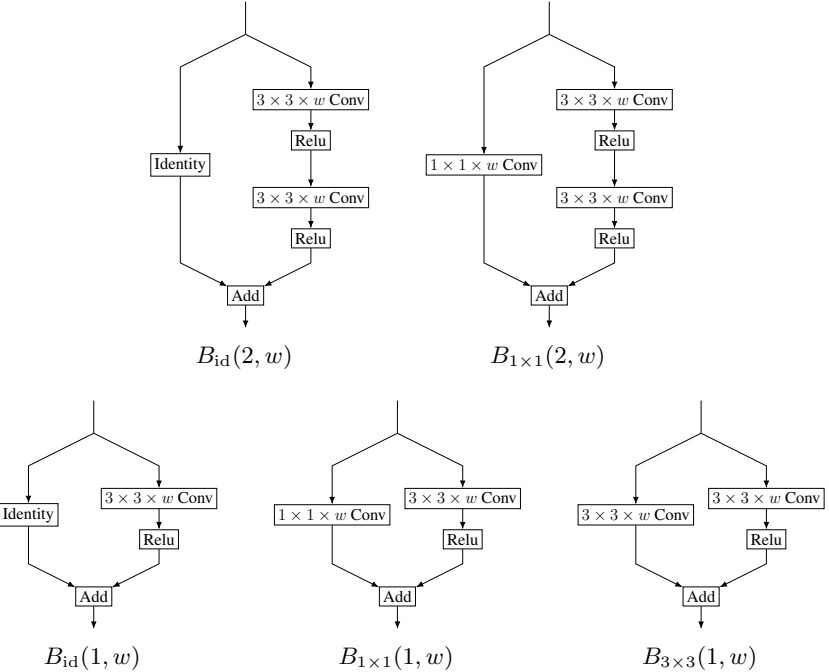

Figure 1: The five tandem blocks used in all of our experiments. The two left-most blocks $B_{\mathrm{id}}(2, w)$ and $B_{\mathrm{id}}(1, w)$ correspond to traditional resnets and the others to more general tandem nets.

Reasonable heuristic arguments could be made to justify all sorts of expectations for the different blocks. For example, one might expect the identity blocks to perform well in deeper networks because they avoid changes in gradient magnitude, but they might also be less effective than blocks that can use $1 \times 1$ convolutions to recombine features in potentially more useful ways. Using $3 \times 3$ convolutions for the linear connections and letting both sides engineer new filters could create more robust networks that need less depth, but it could also be an inefficient use of parameters. Our experiments were designed to find out which of these intuitions are correct.

## 3 EXPERIMENT DESIGN

For our experiments, we built networks of varying widths and depths from our five chosen tandem blocks and tested them on several popular image recognition datasets.

### 3.1 NETWORK ARCHITECTURES

We focused on small architectures ranging from 8 to 26 layers and from 100k to 1.2M parameters. This was appropriate because the goal of these experiments was not to challenge state-of-the-art results produced by much larger models, but rather to compare the five block types in a variety of settings, to gain insight into their relative strengths and weaknesses.

Each experiment features five models, corresponding to the five types of block. To ensure fair comparisons, each model has the same number of layers and nearly the same number of learnable parameters. To achieve the latter, each different type of block must have a different width $w$. In particular, $B_{3\times 3}$ requires significantly smaller values of $w$ than the other blocks because its linear convolutions have many more parameters.

### 3.2 NETWORK SHAPE

For all models we used a simple architecture with three shape hyperparameters. The block layer parameter $l$ sets the number of layers in each block, as in Figure 1. The depth parameter $d$ controls the depth in the network by determining how many times to repeat each block. The width parameter $w$ sets the width of each block and is used to control the number of parameters in each model. The architecture is illustrated in Table 1. In every case, the resulting network had $6d + 2$ layers.

| Component | Repetitions |
|---|---|
| Input | 1 |
| $3 \times 3 \times w$ Conv | 1 |
| $B(l, w)$ | $2d/l$ |
| $B(l, 2w)$ with Stride 2 | 1 |
| $B(l, 2w)$ | $2d/l - 1$ |
| $B(l, 4w)$ with Stride 2 | 1 |
| $B(l, 4w)$ | $2d/l - 1$ |
| Global Average Pooling | 1 |
| Softmax Output Layer | 1 |

Table 1: All of the networks used in our experiments were instances of this meta-architecture. The parameters $w$, $d$ and $l$ are chosen so that the total depth of each network is the same and the total number of parameters is comparable.

### 3.3 REGULARIZATION

In each model, we used both dropout and $L^2$ regularization (weight decay). In all blocks, we applied dropout immediately after the linear and nonlinear sides were added together. In blocks with two nonlinear layers, we also applied dropout after the first nonlinear layer. We applied $L^2$ regularization to the weights of every convolution, both linear and nonlinear, but not to the biases.

We determined the weight decay and dropout rates for each architecture separately for each architecture through a series of grid searches. While different values of these hyperparameters proved optimal for different networks (as measured on a validation set), they were always modest (with dropout values from 0.1 to 0.2 and weight decay values from 0.0001 to 0.0003) and did not significantly change the relative performances of different architectures.

We also used a simple augmentation scheme for the training data: shifting images both horizontally and vertically by a factor of no more than $10\%$ and flipping the images horizontally.

### 3.4 BATCH NORMALIZATION

We tried inserting batch normalization layers in several places in each block—before activations, after activations, before convolutions, after the addition—and were quite surprised to find that none of these approaches was helpful. Contrary to the claims usually associated with this method, we observed that networks with batch normalization achieved about the same performance on average, but were less stable and more sensitive to learning rates. It also took much longer to train networks with batch normalization. Accordingly, we left it out of all of our experiments.

### 3.5 INITIALIZATION

We initialized all of the weights, but not the biases, by sampling from a truncated normal distribution (cut off at two standard deviations) with zero mean. Standard deviations were scaled from a base value by fan-in, as in He et al. (2015). The appropriate base standard deviation varied considerably from network to network. Some were as high as $1.2$ while others were as low as $0.3$. Aside from this variation, the responses to different base standard deviations were consistent. Low values produced networks that didn't learn, while high values produced networks that diverged. For a separate experiment described in Section 5, we tried some non-random initialization schemes to see if networks would learn identity maps on their own.

### 3.6 LEARNING RATE AND DESCENT METHOD

We used the Adam method of gradient descent for all experiments, with the authors' recommended hyperparameters; see Kingma & Ba (2014). Even with adaptive learning rates, we found that a learning rate schedule significantly improved results. For CIFAR-10 and CIFAR-100, we used a learning rate of $0.001$ through epoch 90, then $0.0002$ through epoch 120, and then $0.00004$ until epoch 150. For SVHN and Fashion-MNIST, we scaled this schedule to 100 epoch runs.

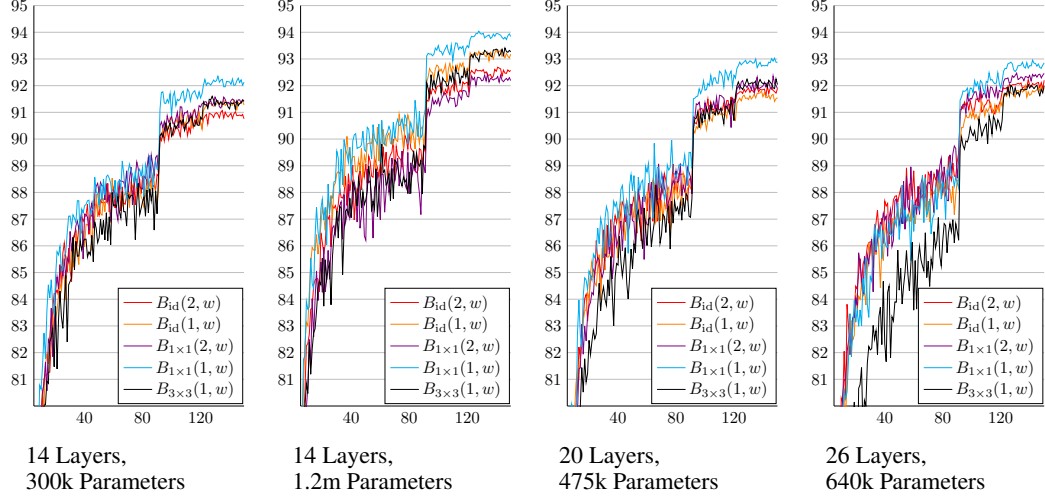

Figure 2: Plots of the test accuracy by epoch from the third-best of each architecture's five runs for all CIFAR-10 experiments. In each case, the tandem model $B_{1\times1}(1, w)$ (light blue) performed best, beating both the resnet models $B_{\text{id}}(1, w)$ and $B_{\text{id}}(2, w)$. The model $B_{1\times1}(1, w)$ was consistently the best in the different runs. Average final accuracy for the five runs is listed in Table 2.

## 4 RESULTS

To evaluate the five blocks, we used each to build several different networks and then tested them on four popular image recognition problems: CIFAR-10, CIFAR-100, SVHN, and Fashion-MNIST. In total, we made the five blocks compete in thirteen challenges. Each challenge was run five times. Table 2 summarizes the results of all of our experiments. To make the graphs in Figures 2, 3, 4, and

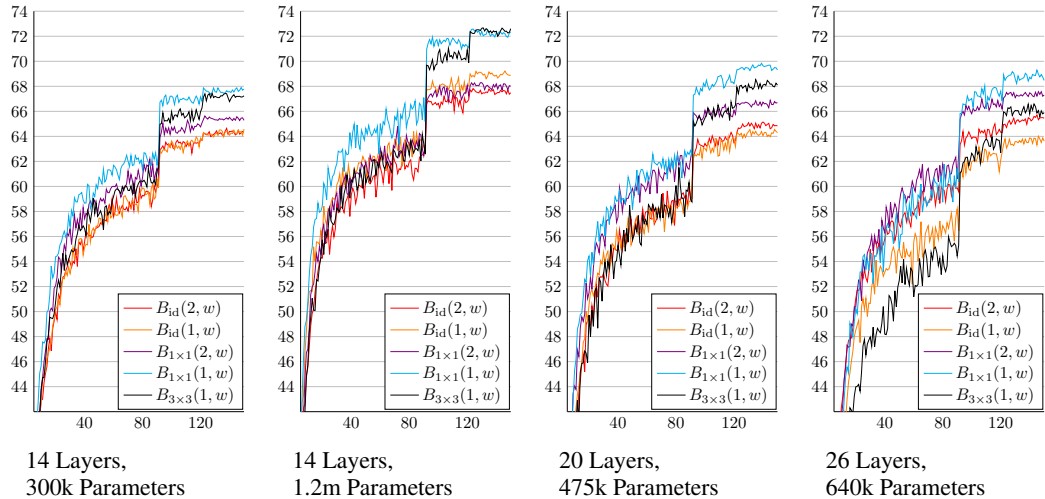

14 Layers,
300k Parameters

14 Layers,
1.2m Parameters

20 Layers,
475k Parameters

26 Layers,
640k Parameters

Figure 3: Plots of the test accuracy by epoch from the third-best of each architecture's five runs for all CIFAR-100 experiments. In all cases the tandem models were clear winners, with $B_{1\times1}(1, w)$ (light blue) performing best or near best each time, and both resnet models $B_{id}(1, w)$ and $B_{id}(2, w)$ at or near the bottom. The tandem model $B_{3\times3}(1, w)$ performed better than the model $B_{1\times1}(2, w)$ in the shallower networks (14 and 20 layers), but $B_{1\times1}(2, w)$ did better in the deeper (26-layer) network. Average final accuracy for the five runs is listed in Table 2.

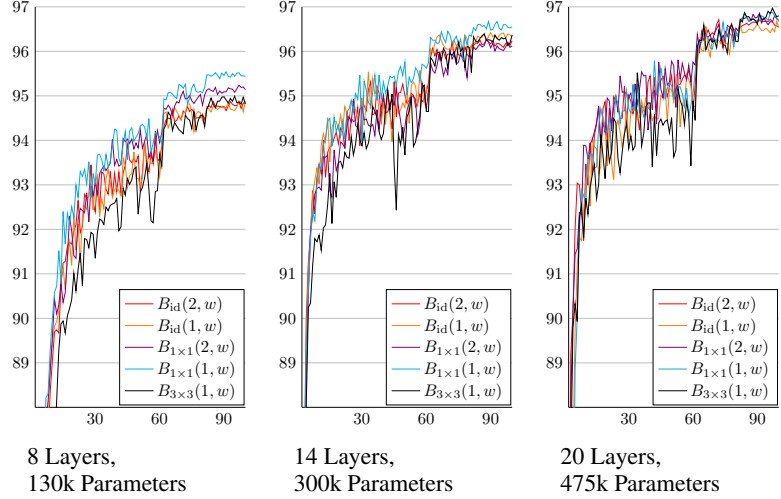

8 Layers,
130k Parameters

14 Layers,
300k Parameters

20 Layers,
475k Parameters

Figure 4: Plots of the test accuracy by epoch from the third-best of each architecture's five runs for all SVHN experiments. Again, in all three cases the tandem models $B_{1\times1}(1, w)$ performed best or near best, and outperformed the resnet models $B_{id}(1, w)$ and $B_{id}(2, w)$, although in the last experiment, all the models—both resnet and more general tandem—have similar performance. Average final accuracy for the five runs is listed in Table 2.

5 represent what these runs actually look like, we plotted the test accuracy from the third-best of each architecture's five runs. The results for standard resblocks were very close to those published by He et al. (2015) and Zagoruyko & Komodakis (2016).

In all four challenges on CIFAR-10 (Figure 2), the $B_{1\times1}(1, w)$ variant (which uses learnable projections instead of identity maps) excelled while the $B_{id}$ variants, with their more standard identity connections, lagged behind.

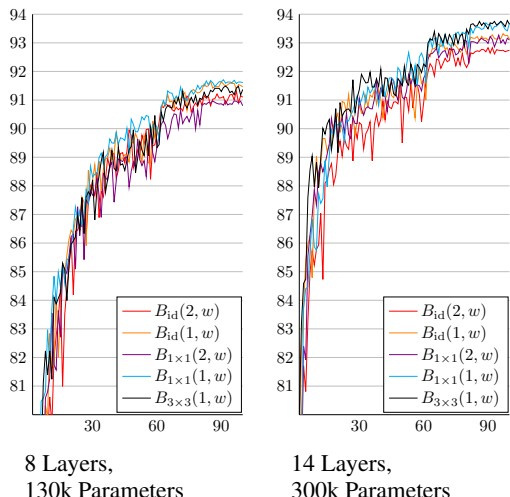

8 Layers,
130k Parameters

14 Layers,
300k Parameters

Figure 5: Plots of the test accuracy by epoch from the third-best of each architecture's five runs for all Fashion-MNIST experiments. Again the tandem model $B_{1\times1}(1,w)$ performed at or near best, although the tandem model $B_{3\times3}(1,w)$ did best in the deeper (14-layer) experiment. The models with two nonlinear layers $B_{id}(2,w)$ and $B_{1\times1}(2,w)$ performed worst. Average final accuracy for the five runs is listed in Table 2.

| Dataset | Layers | Params | $B_{\mathrm{id}}(2)$ | $B_{\mathrm{id}}(1)$ | $B_{1\times1}(2)$ | $B_{1\times1}(1)$ | $B_{3\times3}(1)$ |
|---|---|---|---|---|---|---|---|
| CIFAR-10 | 14 | 300k | 90.91 | 91.30 | 91.43 | **92.14** | 91.44 |
| CIFAR-10 | 14 | 1.2m | 92.48 | 93.12 | 92.18 | **93.83** | 93.24 |
| CIFAR-10 | 20 | 470k | 91.91 | 91.51 | 91.99 | **92.82** | 92.07 |
| CIFAR-10 | 26 | 640k | 92.19 | 91.81 | 92.45 | **92.84** | 91.98 |
| CIFAR-100 | 14 | 300k | 64.28 | 64.66 | 65.31 | **67.74** | 67.25 |
| CIFAR-100 | 14 | 1.2m | 67.31 | 68.87 | 68.13 | 72.34 | **72.69** |
| CIFAR-100 | 20 | 470k | 64.73 | 64.54 | 66.75 | **69.17** | 68.02 |
| CIFAR-100 | 26 | 640k | 65.52 | 63.40 | 67.20 | **68.42** | 65.89 |
| SVHN | 8 | 130k | 94.86 | 94.82 | 95.07 | **95.37** | 94.81 |
| SVHN | 14 | 300k | 96.18 | 96.31 | 96.12 | **96.51** | 96.33 |
| SVHN | 20 | 470k | 96.50 | 96.24 | 96.60 | 96.67 | **96.74** |
| Fashion-MNIST | 8 | 130k | 91.08 | 91.52 | 90.85 | **91.62** | 91.42 |
| Fashion-MNIST | 14 | 300k | 92.74 | 93.23 | 93.16 | **93.71** | 93.65 |

Table 2: The average test accuracy achieved with each tandem block in five repetitions of each experiment. In every case both standard residual blocks $B_{\mathrm{id}}(2,w)$ and $B_{\mathrm{id}}(1,w)$ were outperformed by at least one of the more general tandem blocks. In all but one case (FAshion-MNIST, 8 layers) both of the standard residual blocks were outperformed by at least two of the more general tandem networks.

The results for CIFAR-100 (Figure 3) were similar to those for CIFAR-10 in that the tandem block $B_{1\times1}(1,w)$ performed at or near the top and consistently outperformed all the traditional resnet blocks. The two-layer blocks improved their relative performance as depth increased and $B_{3\times3}$ actually got worse. However, $B_{3\times3}(1,w)$ shined with the mid-depth architectures, particularly on the wider one with 1.2m parameters. This suggests that $B_{3\times3}$ might be even better suited to building relatively wide networks than the resblocks used to achieve state-of-the-art results by Zagoruyko & Komodakis (2016).

On SVHN (Figure 4), the tandem net $B_{1\times1}(1,w)$ once again excelled while the traditional resblocks $B_{\mathrm{id}}(2,w)$ and $B_{\mathrm{id}}(1,w)$ stayed behind. Interestingly, $B_{3\times3}(1,w)$ improved as the networks became deeper, even beating $B_{1\times1}(1,w)$ in the 20-layer network.

On the Fashion-MNIST dataset (Figure 5), $B_{1\times1}(1, w)$ again had the strongest performance, but the resnet $B_{\text{id}}(1, w)$ was not terribly far behind. The tandem block $B_{3\times3}(1, w)$ did better on the deeper network. In these tests the blocks with two nonlinear layers $B_{\text{id}}(2, w)$ and $B_{1\times1}(2, w)$ were significantly behind.

## 5 NON-IDENTITY MAPS

The use of learnable parameters for the linear convolutions in tandem blocks naturally invites several questions. Will linear convolutions with randomly initialized weights learn something similar to an identity map? Will linear convolutions initialized to an identity map stay there, or learn something else? If not identity maps, what do the optimal linear transformations for tandem blocks look like? To explore these questions, we looked at the singular value decompositions of the weight matrices of blocks with $1 \times 1$ linear convolutions. Figure 6 shows that the linear convolutions did not learn identity maps, regardless of initialization scheme.

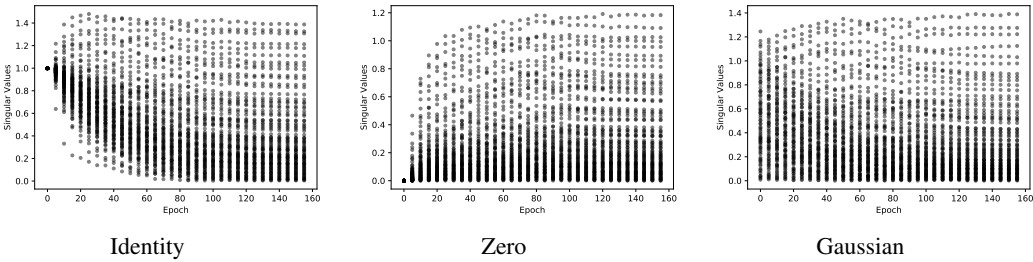

| Identity | Zero | Gaussian |

Figure 6: These plots show the singular values of the weight matrix of the linear $1 \times 1$ convolution in a $B_{1\times1}(1, w)$ block. We tried initializing these weights with identity matrices, zero matrices, and random matrices. All the singular values of the identity matrix are equal to 1, as seen in the initial epoch of the far left panel. Similarly all the singular values for the zero matrix are 0, as seen in the initial epoch of the middle panel. In each case, the network learned a weight matrix that was quite different from any of the initializations. In particular, this shows that identity maps are not even locally optimal in these applications.

## 6 CONCLUSIONS

We generalized residual blocks (which use identity shortcut connections) to tandem blocks (which can learn any linear connection, not just the identity). We found that general linear connections with learnable weights, have the same benefits as the identity maps in residual blocks, and they actually increase performance compared to identity maps. We also showed that linear connections do not learn identity maps, even when initialized with identity weight matrices. These results seem to confirm that the success of residual networks and related architectures is not due to special properties of identity maps, but rather is simply a result of using linear maps to complement nonlinear ones.

The additional flexibility gained by replacing identity maps with convolutions led to better results in every one of our experiments. This was not due to extra parameters, as we adjusted layer widths to keep parameter counts as close to equal as possible. Instead, general linear convolutions appear to do a better job than identity maps of working together with nonlinear convolutions.

Our results further suggest that tandem blocks with a single nonlinear convolution tend to outperform those with two, but blocks that use $3 \times 3$ convolutions for their linear connections may be better in wide networks than those with $1 \times 1$s.

Finally, we note that there are many more possible types of tandem block than those we have considered here, and many more applications in which to test them.

ACKNOWLEDGMENTS

Removed for review.

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
