# OpenReview forum: "Tandem Blocks in Deep Convolutional Neural Networks"
_ICLR.cc/2018/Conference — Reject_

### Official Review · AnonReviewer2 · 2017-11-09
**Well-written, easily digestable, somewhat marginal paper**

**Rating:** 7
**Confidence:** 4

**Review:**

This paper investigates the effect of replacing identity skip connections with trainable convolutional skip connections in ResNet. The authors find that in their experiments, performance improves. Therefore, the power of skip connections is due to their linearity rather than due to the fact that they represent the identity.

Overall, the paper has a clear and simple message and is very readable. The paper contains a good amount of experiments, but in my opinion not quite enough to conclude that identity skip connections are inherently worse. The question is then: how non-trivial is it that tandem networks work? For someone who understands and has worked with ResNet and similar architectures, this is not a surprise. Therefore, the paper is somewhat marginal but, I think, still worth accepting.

Why did you choose a single learning rate for all architectures and datasets instead of choosing the optimal one for each archtitecture and dataset? Was it a question of computational resources? Using custom step sizes would strenghten your experimental results significantly. In the absence of this, I would still ask that you create an appendix where you specify exactly how hyperparameters were chosen.

Other comments:

- "and that it’s easier for a layer to learn from a starting point of keeping things the same (the identity map) than from the zero map" I don't understand this comment. Networks without skip connections are not initialized to the zero map but have nonzero, usually Gaussian, weights.
- in section 2, reason (ii), you seem to imply that it is a good thing if a network behaves as an ensemble of shallower networks. In general, this is a bad thing. Therefore, the fact that ResNet with tandom networks is an ensemble of shallower networks is a reason for why it might perform badly, not well. I would suggest removing reason (ii).
- in section 3, reason (iii), you state that removing nonlinearities from the skip path can improve performance. However, using tandom blocks instead of identity skip connections does not change the number of nonlinearity layers. Therefore, I do not see how reason (iii) applies to tandem networks.
- "The best blocks in each challenge were competitive with the best published results for their numbers of parameters; see Table 2 for the breakdown." What are the best published results? I do not see them in table 2.

---

> ### Author Response · Authors · 2018-01-05
> **Response to AnonReviewer2**
>
> We appreciate the thoughtfulness that went into this review.  We feel that we have substantially improved the paper as a result of this review and the other two.
>
> First, it is important to note that we aren't replacing identity shortcuts so much as generalizing them. Tandem blocks include standard (identity) residual blocks as a special case. An identity shortcut is just a 1x1 convolution without non-linearity whose weight matrix is fixed as an identity matrix. The intent of our paper is to show that the latter property is unnecessary and limiting. The weight matrix of the linear shortcut doesn't need to be fixed (it can be learnable) and it doesn't need to be an identity matrix (either at initialization or after training). The linear convolution doesn't even need to be 1x1, which is particularly surprising. Because a number of notable papers contain assertions to the contrary (that identity connections are necessary and/or optimal), we believe that our contribution is both new and important. However, we failed to clearly express this and have revised the paper accordingly.
>
> Second, it is important to make clear that we didn’t just switch identity connections to linear connections, rather we also reduced the number of neurons per layer in the linear case so that the total number of parameters did not increase in our comparisons.  In other words, we narrowed the layers to make the contests fair.  This wasn’t as clear as it could have been in the initial version of the paper.  We hope it is clearer now.
>
> To address the reviewer’s question about learning rates, we did do a grid search across a number of learning rate schedules, testing them separately for each architecture, and (surprisingly) the same rate schedule turned out to be optimal for every architecture. In Section 3 we clarified our approach.  We also clarified how we performed the searches for dropout and weight decay parameters, which convinced us to use different values for different architectures; see Section 3 for details.
>
> Response to Other Comments:
>
> The comment about learning from the zero map has been clarified to indicate that we initialized weights with small Gaussian values, as is standard practice.
>
> We removed the paragraph about tandem networks acting as ensembles of shallower networks, per the reviewer's suggestion. We removed the paragraph about removing nonlinearities for the same reason and Section 2 is clearer as a result.
>
> We should clarify that our results are competitive with those achieved in other ResNet papers. We mention this primarily to establish that we correctly recreated their architectures for our experiments, making the comparisons fair. Our networks may not beat more complex architectures (such as Inception) on a per-parameter basis, but that isn't the goal. We're only investigating the question of shortcut connections, so we tried not to introduce any extra variables.

---

> > ### Comment · AnonReviewer2 · 2018-01-08
> > **I like the revision**
> >
> > I like the authors thoughtful response to my points and those raised by other reviewers. Also I was not aware that the major ResNet papers took a position in favor of identity connections. I am more convinced than before that this paper is an accept.
> >
> > One note: Regarding the statement that ResNet combats exploding gradients, which one of the other reviewers objected to, this has been demonstrated in "Gradients explode - deep networks are shallow - ResNet explained" as submitted to this conference: https://openreview.net/forum?id=HkpYwMZRb (I hope you'll allow me to be so bold as to shill my own paper :)

---

### Official Review · AnonReviewer1 · 2017-11-27
**Well structured analysis paper on shortcut connections but contributions/results are not compelling**

**Rating:** 5
**Confidence:** 4

**Review:**

This paper performs an analysis of shortcut connections in ResNet-like architectures. The authors hypothesize that the success of shortcut connections comes from the combination of linear and non-linear features at each layer and propose to substitute the identity shortcuts with a convolutional one (without non-linearity). This alternative is referred to as tandem block. Experiments are performed on a variety of image classification tasks such as CIFAR-10, CIFAR-100, SVHN and Fashion MNIST.

The paper is well structured and easy to follow. The main contribution of the paper is the comparison between identity skip connections and skip connections with one convolutional layer.

My main concerns are related to the contribution of the paper and experimental pipeline followed to perform the comparison. First, the idea of having convolutional shortcuts was already explored in the ResNet paper (see https://arxiv.org/pdf/1603.05027.pdf). Second, given Figures 3-4-5-6, it would seem that the authors are monitoring the performance on the test set during training. Moreover, results on Table 2 are reported as the ones with “the highest test accuracy achieved with each tandem block”. Could the authors give more details on how the hyperparameters of the architectures/optimization were chosen and provide more information on how the best results were achieved?

In section 3.5, the authors mention that batchnorm was not useful in their experiments, and was more sensitive to the learning rate value. Do the authors have any explanation/intuition for this behavior?

In section 4, authors claim that their results are competitive with the best published results for a similar number of parameters. It would be beneficial to add the mentioned best performing models in Table 2 to back this statement. Moreover, it seems that in some cases such as SVHN the differences between all the proposed blocks are too minor to draw any strong conclusions. Could those differences be due to, for example, luck in picking the initialization seed? How many times was each experiment run? If more than once, what was the std?

The experiments were performed on relatively shallow networks (8 to 26 layers). I wonder how the conclusions drawn scale to much deeper networks (of 100 layers for example) and on larger datasets such as ImageNet.

Figures 3-5 are not referenced nor discussed in the text.

Following the design of the tandem blocks proposed in the paper, I wonder why the tandem block B3x3(2,w) was not included.

Finally, it might be interesting to initialize the convolutions in the shortcut connections with the identity, and check what they have leant at the end of the training.

Some typos that the authors might want to fix:

- backpropegation -> backpropagation (Introduction, paragraph 3)
- dropout is a kind of regularization as well (Introduction, second to last paragraph)
- nad -> and (Sect 3.1. paragraph 1)

---

> ### Author Response · Authors · 2018-01-05
> **Response to AnonReviewer1**
>
> First, we'd like to clarify what we see as the central thesis of our paper. We aren't replacing identity shortcuts so much as generalizing them. Tandem blocks include standard (identity) residual blocks as a special case. An identity shortcut is just a 1x1 convolution without non-linearity whose weight matrix is fixed as an identity matrix. The intent of our paper is to show that the latter property is unnecessary and limiting. The weight matrix of the linear shortcut doesn't need to be fixed (it can be learnable) and it doesn't need to be an identity matrix (either at initialization or after training). The linear convolution doesn't even need to be 1x1, which is particularly surprising. Because a number of notable papers contain assertions to the contrary (that identity connections are necessary and/or optimal), we believe that our contribution is both new and important. However, we failed to clearly express this and have revised the paper accordingly.  We are grateful to the reviewer for pointing out the weaknesses of the submitted draft.
>
> The linked paper (“Identity Mappings in Deep Residual Networks” by He et al.) does explore the idea of using learnable linear 1x1 convolutions instead of identity mappings, as does the original ResNet paper. Both conclude that identity connections are superior on the grounds that they work better in extremely deep networks because they don't scale gradients. We did not intend to claim to be the first to use linear 1x1s in this way. Instead, our primary aim was to challenge the conclusion that identity connections are superior.  We have now clarified this in the revised paper.
>
> Much of the initial explanation for why identity shortcut connections were important had to do with building extremely deep networks. However, Zagoruyko and Komodakis showed that wider, shallower networks are superior even with traditional resblocks (https://arxiv.org/pdf/1605.07146.pdf). So it's important to ask what types of shortcut connections work best in these cases.
>
> In reading this review, it was clear that we needed to explain more thoroughly our experimental procedures, including our use of train/val/test splits and hyperparameter grid search.  As is traditional, we do not use the test set for hyperparameter selection, but rather a separate validation set.  The test set is only used for final evaluation. We hope this is now clear in the paper.
>
> Unfortunately, we don't have a good explanation for the effects of batch normalization in our experiments. We expected it to help, but this simply wasn't what we observed. This question certainly merits further investigation.
>
> We should clarify that our results are competitive with those achieved in other ResNet papers. We mention this primarily to establish that we correctly recreated their architectures for our experiments, making the comparisons fair. Our networks may not beat more complex architectures (such as Inception) on a per-parameter basis, but that isn't the goal. We're only investigating the question of shortcut connections, so we tried not to introduce any extra variables.
>
> The differences between architectures in some experiments were indeed too small to indicate that one architecture was better than another, and we don't want to imply otherwise. Our goal is to show that non-identity connections were better than identities in some experiments and comparable in others. Both cases contradict the near-universal assertions that identity connections are somehow special or optimal.  It is important to make clear that we didn’t just switch identity connections to linear connections, rather we also reduced the number of neurons per layer so that the total number of parameters did not increase in our comparisons.  In other words, we narrowed the layers to make the contests fair.
>
> We would love to provide results on larger datasets, however, our computational resources are an issue. Testing extremely deep networks would also be interesting, but we would expect to observe the same thing as everyone else—that extremely deep networks take much longer to train and offer at best marginally better performance.
>
> We have referenced and discussed all of the figures explicitly in the revised text.
>
> IMPORTANT:  At the reviewer's suggestion, we confirmed using the singular value decomposition that linear connections with standard initializations (zero mean and small variance) did not learn identity maps and that linear connections initialized to the identity did not stay there.  In other words, these maps are truly non-identity in nature.  This was an excellent suggestion from the reviewer and has (in our opinion) substantially strengthened our argument and the paper.
>
> We noted that dropout is a kind of regularization, this and the typos are fixed.

---

### Official Review · AnonReviewer3 · 2017-11-27
**Weak contribution.**

**Rating:** 4
**Confidence:** 4

**Review:**

The paper is well written, has a good structure and is easy to follow. The paper investigates the importance of having the identity skip connections in residual block. The authors hypothesize that changing the identity mapping into a linear function would be beneficial. The main contribution of the paper is the Tandem Block, that is composed of two paths, linear and nonlinear, the outcome of two paths is summed at the end of the block. Similarly, as for residual blocks in ResNets, one can stack together multiple Tandem Blocks. However, this contribution seems to be rather limited. He at. al. (2016) introduces a Tandem Block like structure, very similar to B_(1x1)(2,w), see Fig. 2(e) in He at. al. (2016). Moreover, He et. al (2016) shows in Tab 1 that for a ResNet 101 this tandem like structure performs significantly worse than identity skip connections. This should be properly mentioned, discussed and reflected in the contributions of the paper.

Result section:
My main concern is that it seems that the comparison of different Tandem Blocks designs has been performed on test set (e. g. Table 2 displays the highest test accuracies) . Figs 3, 4, 5 and 6 together with Tab. 2 monitors test set. The architectural search together with hyperparameters selection should be performed on validation set.


Other issues:
- Section 1: “… ResNets have overcome the challenging technical obstacles of vanishing/exploding gradients… “. It is clear how ResNet address the issue of vanishing gradients, however, I’m not sure if ResNet can also address the problem of exploding gradients. Can authors provide reference for this statement?
- Experiments: The authors show that on small size networks Tandem Block outperforms Residual Blocks, since He at. al. (2016) in Tab 1 showed a contrary effect, does it mean that the observations do not scale to higher capacity networks? Could the authors comment on that?

---

> ### Author Response · Authors · 2018-01-05
> **Response to AnonReviewer3**
>
> We appreciate the thoughtfulness that went into this review.  We feel that we have substantially improved the paper as a result of this review and the other two.
>
> Following the reviewers comments, we have clarified that we aren't contrasting residual blocks with tandem blocks. It is more accurate to say that tandem blocks generalize residual blocks, including identity connections as a special case.
>
> The paper “Identity Mappings in Deep Residual Networks” by He et al does explore the idea of using learnable linear 1x1 convolutions instead of identity mappings, as does the original ResNet paper. Both conclude that identity connections are superior on the grounds that they work better in extremely deep networks because they don't scale gradients. We did not intend to claim to be the first to use linear 1x1s in this way. Instead, our primary aim was to challenge the conclusion that identity connections are superior. We have now clarified this and discussed the relevant papers in the revised paper.
>
> Much of the initial explanation for why identity shortcut connections were important had to do with building extremely deep networks. However, Zagoruyko and Komodakis showed that wider, shallower networks are superior even with traditional resblocks (https://arxiv.org/pdf/1605.07146.pdf). So it's important to ask what types of shortcut connections work best in these cases. Our experiments show that learnable linear connections are as good as or better than identity connections in networks of practical size.
>
> In reading this review, it was clear that we needed to explain more thoroughly our experimental procedures, including our use of train/val/test splits and hyperparameter grid search.   As is traditional, we do not use the test set for hyperparameter selection, but rather a separate validation set.  The test set is only used for final evaluation. We hope this is now clear in the paper.
>
> We have fixed an incorrect statement to reflect the fact that identity connections don't prevent exploding gradients.  We thank the reviewer for calling that to our attention.
>
> It is important to differentiate between network capacity and network depth. Zagoruyko and Komodakis used networks of tremendous capacity (but not particularly great depth) and outperformed the original ResNets which were much deeper. We would love to provide results for much larger networks (in terms of parameter count) and also on larger datasets. However, our computational resources are an issue. Testing extremely deep networks would also be interesting, but we would expect to observe the same thing as everyone else—that extremely deep networks take much longer to train and offer at best marginally better performance.

---

### Comment · AnonReviewer2 · 2017-11-10
**Can you see my review?**

Dear authors,

I posted my review recently. I am curious: Can you see the review? Because when I log out of my account, I can no longer see it. Hence, the review is (so far) not public. I am wondering whether at least you can see it.

Thanks,

---

> ### Author Response · Authors · 2017-11-12
> **Your review didn't post, but did come as an email.**
>
> Dear Reviewer,
>
> We can't see your review on OpenReview, but we did receive it via email. We appreciate your analysis and look forward to answering your questions and making the appropriate revisions during the discussion period. Hopefully the review will post soon.
>
> Thanks,
> The Authors

---

### Author Response · Authors · 2018-01-05
**Summary of Changes**

In response to the reviewers' comments, we have made a number of improvements to the paper. Most importantly, we:

- Clarified that the primary purpose of the paper was not to introduce novel architectures, but to challenge the conclusion that identity shortcuts are superior to other linear shortcuts. We show experimentally that this is not the case for any of the networks we trained.

- Added a small section (based on a reviewer's suggestion) showing that the linear connections in our networks did not learn to imitate identity connections, even if they were initialized with identity weight matrices. This supports the conclusion that learnable weights add real value to linear connections.

- Explained that we used validation data to determine hyperparameters and test data only for our final comparisons between architectures, following the standard practice. We also clarified that we were comparing average performance across series of five runs for each experiment. Both points were unclear in our original submission.

- Removed some introductory comments that were confusing or distracted from our main points.

- Stressed that differences in performance were not due to some architectures having unfair advantages due to greater numbers of parameters. We were careful to keep parameter counts as close as possible by adjusting layer widths separately for each architecture.

- Discussed the relevant literature more thoroughly in the first two sections of the paper.

We also made a number of minor corrections and clarifications.

---

### Public Comment · ~Oshrat_Bar1 · 2018-01-07
**ICLR 2018 Reproducibility Challenge - Questions**

Hi,
As a final project in deep learning seminar in Tel Aviv University, I am reproducing the experiments described in your paper.
I’ll much appreciate it if you can answer a few questions regarding the implementation details.

1. Tandem blocks with stride 2
I assume stride 2 in the spatial dimensions.
As I understand, on those blocks, the linear layer is either 1x1 convolution or 3x3 convolution (not identity) as the input and output differs in the third dimension.
Is there a stride 2 in the linear part as well?
In blocks with l=2, there are 2 nonlinear layers (3x3 convolution) and only 1 linear layer (1x1 convolution). Assuming stride 2 in all the 3 convolutions, linear output and nonlinear output have different dimensions and can’t be summed together. How do you cope with that?

2. Initialization
In the paper, you mentioned that the initialization was done as in He et al. (2015).
As I understand, std in layer l is sqrt( 2/nl), while nl is the number of the kernel parameter.
You also mentioned a “base standard deviation” that varied considerably from network to network.
How does the “base standard deviation” affect the std computation?
What “base standard deviation” did you use in each model?
What is the std used for the softmax output layer weights?
Is it correct that you initialized biases to 0?

3. Training
What was the batch size that you used in each of the experiments?
You mentioned a use of validation set. What portion of the training set was used for training in each of the experiments?

4. Regularization
Please specify the weight decay and dropout rates that you used for each of the architectures.

5. Data Augmentation
Please specify the details and amount of the augmentation that you used.

Thanks,
Oshrat Bar
Tel Aviv University
oshratbar@mail.tau.ac.il

---

> ### Author Response · Authors · 2018-01-09
> **Reproducibility Challenge - Answers (Part 1)**
>
> We'd be more than happy to help you recreate our experiments. Hopefully the following will answer most of your questions. We'll also provide a follow-up later this week with code samples and tables of hyperparameter values.
>
> 1. We do mean stride in the usual (spatial) sense. Much like pooling, it reduces the height and width of our image channels. When using identity connections or 1x1 convolutions, using stride 2 simply amounts to taking the sub-image of pixels with even coordinates.
>
> As you observe, this needs to happen on both the linear and nonlinear sides of a block so that they can be added together. When there are two nonlinear layers in a block, we only change the stride on the first one.
>
> 2. The specific initialization method we used was the 'variance scaling' method from the keras package, which uses a standard deviation of sqrt(scale/n) where n is the number of inputs to the layer (which is just the width of the previous layer). We determined the scale parameter experimentally, so we'll have to put together a list of the ones we ended up using for each experiment.
>
> Exact values don't seem to be very important in this case. They just need to be large enough to get the network learning, but not so large that it becomes unstable. All of our values were between 0.3 and 1.2.
>
> 3. We used a batch size of 125 for all experiments.
>
> All of the data sets we used come with designated test sets that we used as such.
>
> For our hyperparameter grid searches, we used 20% of the given training data as validation data and the remaining 80% as training data. We made sure that classes were equally represented in both the training and validation sets.
>
> 4. Our weight decay and dropout values were a little different in each experiment, as dictated by our grid search. We'll make tables of these for you and get them to you soon. However, you may also want to perform your own grid searches for these values. We tried weight decay values from 0.0000 to 0.0004 and dropout rates from 0.0 to 0.3 for each experiment.
>
> 5. For data augmentation, we only used shifts (both vertical and horizontal) and flips (only horizontal). The shifts were limited to 10% of image height/width, so 3.2 pixels.

---

> ### Author Response · Authors · 2018-01-12
> **Reproducibility Challenge - Answers (Part 2)**
>
> The code and hyperparameters we used are now available at https://github.com/tandemblock/iclr2018
>
> We hope this will further clarify our methods and make it easy to reproduce our results.

---

### Author Response · Authors · 2018-01-24
**Experimental Scale**

We appreciate that all of our reviewers responded to our updated paper and we are pleased to see that we managed to address nearly all of their questions and concerns. The only remaining criticisms regard the size of the networks in our experiments. While we were careful to recreate the meta-architecture of Zagoruyko and Komodakis to ensure the most direct and relevant comparisons possible, we understand the desire to see the same experiments done on a larger scale. We did as much as we could in this regard with the financial and computational resources available to us. We believe that our results and analysis make a compelling case for questioning the conventional wisdom in this area and motivate further (and larger) experiments.

---

### Decision · Program_Chairs · 2018-01-29
**ICLR 2018 Conference Acceptance Decision**

**Decision:**

Reject

**Comment:**

The paper presents a good analysis on the use of different linear maps instead of identity shortcuts for resnet.
It is interesting to the community but the experimental justification is insufficient.
1) As pointed out by the reviewer that this work shows "that on small size networks Tandem Block outperforms Residual Blocks, since He at. al. (2016) in Tab 1 showed a contrary effect, does it mean that the observations do not scale to higher capacity networks?", the paper would be much stronger if with experiments justify this claim.
2) "extremely deep networks take much longer to train" is not a valid reason to not conduct such exps.